# High-Precision Iterative Preconditioned Gauss–Seidel Detection Algorithm for Massive MIMO Systems

Mushtaq Ahmad [ID], Xiaofei Zhang *, Imran A. Khoso, Xinlei Shi [ID] and Yang Qian

College of Electronic and Information Engineering, Nanjing University of Aeronautics and Astronautics, Nanjing 211106, China
* Correspondence: zhangxiaofei@nuaa.edu.cn

**Abstract:** Signal detection is a serious challenge for uplink massive multiple-input multiple-output (MIMO) systems. The traditional linear minimum-mean-squared error (MMSE) achieves good detection performance for such systems, but involves matrix inversion, which is computationally expensive due to a large number of antennas. Thus, several iterative methods such as Gauss–Seidel (GS) have been studied to avoid the direct matrix inversion required in the MMSE. In this paper, we improve the GS iteration in order to enhance the detection performance of massive MIMO systems with a large loading factor. By exploiting the property of massive MIMO systems, we introduce a novel initialization strategy to render a quick start for the proposed algorithm. While maintaining the same accuracy of the designed detector, the computing load is further reduced by initialization approximation. In addition, an effective preconditioner is proposed that efficiently transforms the original GS iteration into a new one that has the same solution, but a faster convergence rate than that of the original GS. Numerical results show that the proposed algorithm is superior in terms of complexity and performance than state-of-the-art detectors. Moreover, it exhibits identical error performance to that of the linear MMSE with one-order-less complexity.

**Keywords:** massive MIMO; linear MMSE; signal detection; iterative methods; low-complexity

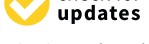



## 1. Introduction

Wireless communications technology has lately seen a remarkable growth in terms of supporting the large amount of mobile users and offering high throughput, with the next generation of cellular networks trying to support exceptional data rates [1], large IoT networks [2], and massive machine-to-machine communications [3]. Moreover, modern wireless communication systems require having high reliability, high energy and spectral efficiency, and high transmission capacity. In order to meet the demands, massive multiple-input multiple-output (MIMO) technology has been applied in the Fifth-Generation (5G) cellular network to manage the limited spectrum resource. It has been considered as a key technology to meet the requirements of high data rates for 5G and beyond wireless systems [4–6]. A large number of antennas at the base station (BS) are employed to serve a relatively small number of user terminals in massive MIMO. By equipping a large number of antennas, more degrees of freedom can be obtained in the wireless channel to simultaneously accommodate more information data, which offer greatly improved energy efficiency and spectral efficiency and provide better reliability compared with the conventional small-scale MIMO systems [7–10]. However, these advantages of massive MIMO systems (over small-scale MIMO) come at the cost of considerably increased computational burden at the BS. The numerous data symbols transmitted by different user terminals undergo multipath and undesired copies of the data symbols coming from different directions of arrival [11,12], and different delays are combined with the direct signal at the receiver side, which corrupts the received symbols. Thus, one of the most computationally intensive tasks is the detection of symbols in the uplink (user transmits to the BS), since the presence of several antennas

needs detection techniques that scale favorably to higher dimensions. The receiver at the BS observes a linear superposition of the independently transmitted information bits, and the task of the signal detection technique is to separate those transmitted information bits. As the number of antennas at the BS grows, the complexity of the detection process increases exponentially. Therefore, the detection process becomes very complex in massive MIMO systems.

The traditional maximum likelihood (ML) detection method [13] can obtain optimality via minimizing the probability of detection error, but computational cost scales exponentially with the number of transmit antennas, which is, hence, computationally prohibitive for large multi-antenna systems. The k-best detection method [14] and the sphere decoding (SD) method [15] are two variants of ML detectors, which balance error performance and computational complexity by controlling the number of nodes in every search phase. Nonetheless, the QR decomposition in these nonlinear detection schemes can lead to low parallelism and high computational cost because of the inclusion of matrix operations such as element elimination. Therefore, to cope with the complexity issue, researchers have considered suboptimal linear detection methods, such as the linear minimum-mean-squared error (MMSE) detector [16], which is computationally less expensive, and it has shown good performance for massive MIMO systems, in particular for a favorable propagation environment and a large loading ratio (M/K) [4], where M and K denote the number of receive and transmit antennas, respectively. It is considered near-optimal for massive MIMO and occupies the benchmark place for most linear iterative detectors. However, the linear MMSE detector needs to compute the inverse of a matrix, and the complexity of matrix inversion increases cubically with the number of users. In other words, the matrices involved in the MMSE become large in dimension for large MIMO systems, and as a result, obtaining the inverse of such high-dimensional matrices is computationally expensive as it increases the cost of receiver development and introduces a considerable delay in processing.

To cut the overhead of high-dimensional matrix inversion while achieving near-MMSE error rate performance, recent works have looked into approximate or implicit matrix inversion methods. The Neumann-series-based detection, which replaces the matrix inversion by either matrix–matrix multiplications or matrix vector multiplications, was developed in [17–19]. It reduces the complexity to some extent, but for $N_{iter} \geq 3$ ($N_{iter}$ shows the number of iterations), it has even higher complexity than the exact matrix inversion method. Newton-iteration (NI)-based [20] detection was proposed to speed up the convergence. Nevertheless, the main disadvantage of the NI method is the same as that of the Neumann method. That is, its computational complexity is higher compared to the exact matrix inversion for more iterations. To further reduce the computational cost, numerous implicit methods such as the Gauss–Seidel (GS) detector [21–24], Jacobi method [25], Richardson iteration [26,27], accelerated over-relaxation (AOR) [28,29], symmetric successive over-relaxation (SSOR) [30], the Lanczos-method-based detector [31], and the conjugate gradient detector [32] have been introduced. These methods compute the estimates of the transmitted symbol without ever computing the matrix inverse. Moreover, in [33], an efficient initialization technique for uplink massive MIMO linear detection methods was developed. The main purpose was to overcome the problem of computing the Gram matrix and match the filter output vector in a pre-processing phase. Although the aforementioned iterative detection schemes are able to realize near-MMSE performance with relatively less computations, they lack the consistency of maintaining a good error rate performance when the number of users grows. Thus, it is very important to develop new detection algorithms to realize a practical receiver for the massive MIMO system with acceptable computational complexity.

## 1.1. Contributions

The main objective of this research is to solve the linear MMSE detection problem for massive MIMO systems using a low-complexity iterative algorithm. To this end, an enhanced version of the GS-based MMSE algorithm is proposed, which replaces the direct

matrix inversion by matrix–vector multiplications. Therefore, unlike the complexity of traditional MMSE method, which is approximately proportional to the cube of the number of users, the complexity of the proposed algorithm is approximately proportional to the square of the number of users for the worst-case scenario. We analyze the property of massive MIMO, and based on that analysis, a novel initializer is proposed. It is then approximated by exploiting the channel hardening property of massive MIMO systems to further reduce the computational load. The proposed initializer achieves the error performance of the conventional diagonal-based initializer with significantly reduced computations. In order to further accelerate the convergence rate of the proposed algorithm, we introduce an efficient preconditioner, which reduces the condition number of the coefficient matrix. The preconditioner converts the original linear system into an equivalent one with the same solution, but a better convergence rate. Computational complexity analysis is presented and simulation results are provided to numerically validate the superiority of the proposed detection algorithm.

Our results demonstrate that the proposed approach outperforms the conventional GS-based detectors for large loading factors and substantially reduces the complexity of the linear MMSE without sacrificing the error performance.

### 1.2. Paper Outline

The remainder of the article is structured as follows: Section 2 details the massive MIMO system model and discusses linear MMSE detection. In Section 3, a low-complexity approach for estimating the transmitted information is presented. Additionally, Section 3 describes the proposed initial solution and develops an efficient preconditioning technique. Simulation results and the analysis of the results are demonstrated in Section 4. Additionally, this section computes and analyzes the computational complexity of the proposed approach and compares it with the traditional massive MIMO detection approaches. Finally, the conclusions are drawn in Section 5.

### 1.3. Notation

Throughout this article, lowercase and bold uppercase letters denote column vectors and matrices, respectively. The $K \times K$ identity matrix is represented by $\mathbf{I}_K$. We denote the inverse and Hermitian transpose, respectively, by $(.)^{-1}$ and $(.)^H$. The vector $\mathbf{a}$ in the $i$th iteration is denoted by $\mathbf{a}^{(i)}$. $a_n$ is the $i$th entry of vector $\mathbf{a}$, and for the element in the $n$th row and $m$th column of matrix $\mathbf{A}$, we use $A_{n,m}$.

## 2. Massive MIMO System Model and Signal Detection

We considered an uncoded uplink massive MIMO system with $M$ active antennas at the BS and serving $K$ single-antenna users simultaneously, as shown in the Figure 1. Usually, the number of antennas at the BS is much larger than the number of users in massive MIMO systems. Suppose the transmitted symbol sent from the $m$th user is denoted as $x_m(1 \leq m \leq K)$, which comprises $J(= \log_2 Q)$ bits per symbol, which is generated from a Q-ary constellation $\mathcal{M}$ with $\sum_{x \in \mathcal{X}} x = 0$ and $\sum_{x \in \mathcal{X}} |x|^2 = Q$. The generated data are mapped and then demultiplexed into K separate independent bit streams, which results in the transmitted vector. For different users, the bit streams are intended to be simultaneously sent to the BS. Thus, the transmitted data stream vector of K users is denoted by $\mathbf{x} = [x_1, x_2, \ldots, x_K]^T$. Then, the standard input–output relation to model a MIMO wireless channel can be expressed as [34]

$$\mathbf{y} = \mathbf{Hx} + \mathbf{n}, \tag{1}$$

where $\mathbf{y} = [y_1, y_2, \ldots, y_M]$ is the $K \times 1$ received symbol vector and $\mathbf{H} = [h_1, h_2, \ldots, h_K]^T$ denotes the $M \times K$ flat Rayleigh fading channel matrix, where the $m(1 \leq m \leq K)$th column vector $h_m$ designates the channel response between the $m$th transmit antenna and all receiving active antennas. Moreover, $\mathbf{n}$ shows the additive white Gaussian noise

(AWGN) vector with independent mean zero components with $\sigma^2$ being the variance. In this circumstance, the average received signal-to-noise ratio (SNR) can be computed as $K/\sigma^2$. We considered, for simplicity, that the channel state information at the receiver is perfectly known.

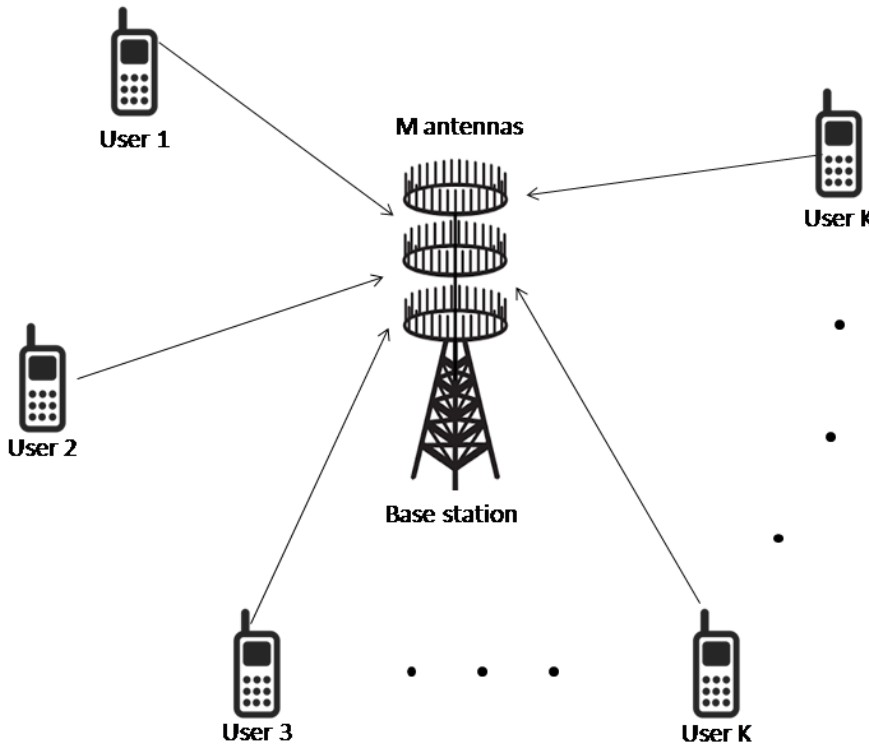

**Figure 1.** Uplink massive MIMO system with a BS employing M antennas and simultaneously serving K users.

The information bits transmitted by various users to the BS overlap and characteristically result in multiuser interference at the receiver in the multiuser uplink large MIMO systems. The multiuser signal detector performs the task of estimating the the transmitted signal vector at the BS from the noisy received signal vector. The signal estimation at the BS employing linear MMSE criteria is given as [35]

$$\hat{\mathbf{x}} = \left( \mathbf{H}^H\mathbf{H} + \frac{\sigma^2}{E_x}\mathbf{I_K} \right)^{-1} \mathbf{H}^H\mathbf{y} = \mathbf{A}^{-1}\hat{\mathbf{y}}, \tag{2}$$

where $E_x$ is the average symbol energy and $\mathbf{A}$ is the regularized Gram matrix (or MMSE filtering matrix), which can be described as

$$\mathbf{A} = \mathbf{H}^H\mathbf{H} + \frac{\sigma^2}{E_x}\mathbf{I_K} = \mathbf{G} + \frac{\sigma^2}{E_x}\mathbf{I_K}, \tag{3}$$

where $\mathbf{G}$ is the Gram matrix. The vector $\hat{\mathbf{y}}$ in (2) denotes the matched filter vector, and it is given by

$$\hat{\mathbf{y}} = \mathbf{H}^H\mathbf{y}. \tag{4}$$

The underlying idea of linear MMSE detector (2) is to invert the effect of the MIMO channel matrix. The matrix inversion involved in the MMSE detector makes it challenging since it entails cubic computational complexity with respect to the number of user terminals, which eventually restricts the possible application in future large wireless systems such as the beyond 5G and Sixth-Generation (6G) systems. It can easily be observed that finding the

solution of the linear MMSE problem is nothing but solving a set of linear equations given by $\mathbf{Ax} = \mathbf{b}$. Hence, numerous alternate methods that do not require the matrix inversion, such as the GS method, have successfully been utilized for massive MIMO detection.

## 3. Proposed Algorithm

It was discussed in Section 2 that, for massive MIMO systems, the linear MMSE detection algorithm can realize good performance. However, a large number of users and antennas at the BS increase the computational burden of MIMO detection by orders of magnitude. Unlike conventional MIMO, in massive MIMO systems, the channel hardening phenomenon can be exploited due to a large number of antennas to cancel the characteristics of a small-scale fading [4]. In this phenomenon, as the number of antennas increases, the variance of the mutual information of the MIMO channel grows very slowly relative to its mean or even shrinks [36]. As the number of transmit and receive antennas increase while keeping their ratio unchanged, the singular-value distribution of the MIMO channel matrix turns out to be less sensitive to the actual distribution of the entries of the channel matrix [36], which is due to the Marchenko–Pastur theorem [37]. Channel hardening can be observed in a system when [38]

$$\frac{\|\mathbf{h}_{mk}\|^2}{E[\|\mathbf{h}_{mk}\|^2]} \to 1, \tag{5}$$

almost surely as $M \to \infty$. Equation (5) states that the gain $\|\mathbf{h}_{mk}\|^2$ of an arbitrary fading channel $\mathbf{h}_{mk}$ is close to its mean value when there are many antennas.

The interesting characteristic in this phenomenon is that it becomes more dominant when the number of receive antennas is much greater that the number of transmit antennas. Furthermore, the MMSE filtering matrix is Hermitian positive definite for massive MIMO systems, and it was shown in [4] that each entry of the diagonal component converges to a fixed value $M$. This is due to the fact that, when the number of receive antennas is very large compared to the number of users, the channel matrix is asymptotically orthogonal [4]. Let $\mathbf{d}$ denote the zero vector; considering the fact that the components of full-rank matrix $\mathbf{H}$ are i.i.d. random variables, then $\mathbf{Hd} = 0$, and for arbitrary non-zero vector $\mathbf{f}$,

$$(\mathbf{Hf})^H \mathbf{Hf} = \mathbf{f}^H (\mathbf{H}^H \mathbf{H}) \mathbf{f} > 0. \tag{6}$$

Moreover,

$$\mathbf{G} = \mathbf{H}^H \mathbf{H} = (\mathbf{H}^H \mathbf{H})^H = \mathbf{G}^H. \tag{7}$$

Hence, the MMSE filtering matrix is positive definite and Hermitian. Using these properties of large MIMO systems, iterative approaches can be applied to compute the approximate solution with significantly lower complexity. Consequently, various approximate iterative detection algorithms with low complexity are being developed or improved to achieve near-optimal error rate performance. Among these detection algorithms, GS-based detection [21] achieves good detection accuracy. We further improved the performance of the conventional GS and reduced the computational complexity in this work.

Consider the linear system:

$$\mathbf{Au} = \mathbf{b}, \tag{8}$$

where $\mathbf{A}$ is a square matrix and $\mathbf{u}$ and $\mathbf{b}$ are $K \times 1$ vectors. Equation (8) is equivalent to the MMSE problem (2). Since it is computationally expensive to solve (8) directly, we applied the GS method to solve it iteratively. As previously mentioned, the MMSE filtering matrix $\mathbf{A}$ is Hermitian positive definite for massive MIMO systems, and we can decompose it as

$$\mathbf{A} = \mathbf{D} + \mathbf{L} + \mathbf{L}^H, \tag{9}$$

where $\mathbf{D}$, $\mathbf{L}$, and $\mathbf{L}^H$, respectively, stand for the diagonal part and the strictly lower and strictly upper triangular parts of $\mathbf{A}$. Then, to reconstruct the transmitted signal vector, the GS iteration can be expressed as

$$
\mathbf{x}_i^{(k)} = \frac{1}{\mathbf{A}_{ii}} \left( \hat{\mathbf{y}}_i - \sum_{j<i} \mathbf{A}_{ij} \mathbf{x}_j^{(k)} - \sum_{j>i} \mathbf{A}_{ij} \mathbf{x}_j^{(k-1)} \right),
$$

$$
i, j = 1, 2, \ldots, N, \tag{10}
$$

where $\mathbf{x}^{(0)}$ is an arbitrary initial vector, $\mathbf{A}_{ij}$ denotes the entry of $\mathbf{A}$ in the $i$th row and $j$th column, and $\hat{\mathbf{y}}_i$, $\mathbf{x}_i^{(k)}$, and $\mathbf{x}_i^{(k-1)}$ represent the $i$th entry of the received symbol vector $\hat{\mathbf{y}}$ and transmitted symbol vectors $\mathbf{x}^{(k)}$ and $\mathbf{x}^{(k-1)}$, respectively. The GS method was applied in [21] to detect the signal vector. A new initializer and efficient preconditioning technique are proposed in this section to make the GS method applicable in practical massive MIMO scenarios.

*3.1. Proposed Initialization*

A proper initialization can lead to a faster convergence and affect both the detection accuracy and complexity of the final solution. Iterative methods usually use a zero vector as the initialization, which requires more iterations to realize the final estimation. The computational cost of each iteration in massive MIMO systems is very high due to a large number of antenna elements. In addition, the conventional GS detector uses a diagonal component as the initialization. Though it obtains better results, this is at the cost of increased computational burden. Hence, finding the optimal solution with less iterations is crucial to implementing massive MIMO.

According to the random matrix theory, i.e., the Marchenko–Pasture theorem, when each component of the matrix channel $\mathbf{H}$ is independently and identically distributed at zero mean, the ratio of the two tends to a constant ($M/K \rightarrow$ constant), and the number of columns and the number of rows tend to infinity, i.e., $M, K \rightarrow \infty$, the off-diagonal entries tend to zero, and the diagonal component of the matrix $\mathbf{H}^H \mathbf{H}$ tend to a certain constant. Thus, for massive MIMO systems, all diagonal components of $\mathbf{H}^H \mathbf{H}$ are positive and [4]

$$
\mathbf{H}^H \mathbf{H} \approx M\mathbf{I}, \tag{11}
$$

and the eigenvalues of $\mathbf{A}$ converge to a fixed deterministic distribution [4]. Inspired by this, the matrix $\mathbf{A}$ can be approximated as

$$
\mathbf{A}_{i,j} = \begin{cases} \lambda_{max}, & i = j, \\ 0, & i \neq j, \end{cases} \tag{12}
$$

where $\lambda_{max}$ is the maximum eigenvalue of the MMSE filtering matrix. (12) shows that each entry of $\mathbf{A}$ is approximately equal to $\lambda_{max}$. Based on the above analysis, we propose a low-complexity initialization given as follows:

$$
\mathbf{x}^{(0)} = \frac{1}{\lambda_{max}} \hat{\mathbf{y}}. \tag{13}
$$

The proposed initialization technique significantly accelerates the convergence rate compared to conventional zero vector initialization and achieves the desired detection performance with few iterations, which reduces the complexity of the proposed detector significantly. Note, however, that the proposed initializer depends on $\lambda_{max}$, which is difficult to determine in practice. However, since the elements of $\mathbf{H}$ are i.i.d. complex

Gaussian random variables, $\mathbf{H}^H\mathbf{H}$ is a complex central Wishart matrix. Hence, as $M$ increases, the largest eigenvalue of $\mathbf{A}$ converges to a deterministic value [4]:

$$\hat{\lambda}_{max} = M\left(1 + \sqrt{\frac{M}{K}}\right)^2,\tag{14}$$

and from (14), it can be noted that the proposed initializer only depends on the system parameters.

### 3.2. Proposed Preconditioning Technique

Compared to direct approaches, iterative approaches often need fewer operations, especially when an approximate solution provides good accuracy. However, iterative techniques have degraded performance, and preconditioning is necessary in order to achieve convergence within few iterations. For challenging problems in scientific computation, it is generally known that preconditioning is the most important ingredient in the design of efficient solvers. The preconditioning techniques are used for transforming the system into another system (preconditioning system) that has more favorable properties for the iterative solution. The rate of convergence of many iterative methods depends inversely on the condition number of the coefficient matrix. If the spectral condition number is large, the asymptotic approximation demonstrates that the convergence is slow. If the spectral condition number is of moderate size, a moderate convergence speed results. If, however, the spectral condition number is very close to 1, we have very fast convergence [39]. In general, preconditioning, when applied to an iterative method, improves the spectral properties of the coefficient matrix, i.e., minimizes the condition number, thereby maximizing the convergence of the iterative method [40]. The idea to minimize the condition number and, hence, maximize the convergence rate by applying a preconditioning technique is shown to be computationally feasible. Since the preconditioner acts on the spectral radius of the iteration matrix, it would be useful to choose an optimal preconditioner for a given linear system, that is a preconditioner that is able to achieve the required convergence with fewer iterations.

If we split the matrix $\mathbf{A} = \mathbf{M} - \mathbf{N}$ with a nonsingular matrix $\mathbf{M}$, then the basic iterative method based on (8) can be expressed as

$$\mathbf{M}\mathbf{x}^{(k+1)} = \mathbf{N}\mathbf{x}^{(k)} + \mathbf{b},\tag{15}$$

We can also write (15) as follows:

$$\mathbf{x}^{(k+1)} = \mathbf{B}\mathbf{x}^{(k)} + \mathbf{c},\tag{16}$$

where $\mathbf{B} = \mathbf{M}^{-1}\mathbf{N}$ and $\mathbf{c} = \mathbf{M}^{-1}\mathbf{b}$.

We assume, for simplicity, that the matrix $\mathbf{A}$ has unit diagonal entries, and let

$$\mathbf{A} = \mathbf{I} - \mathbf{L} - \mathbf{U},\tag{17}$$

where $\mathbf{U}$ and $\mathbf{L}$ are the upper triangular part and strictly lower triangular part of $\mathbf{A}$, respectively. Then, the iteration matrix of the classical GS scheme is given by $\mathbf{B} = (\mathbf{I} - \mathbf{L})^{-1}\mathbf{U}$.

In order to improve the convergence properties of the classical GS detection method, we transformed the original linear system (8) into the preconditioned linear system by multiplying both sides with a nonsingular matrix $\mathbf{T}$:

$$\mathbf{T}\mathbf{A}\mathbf{x} = \mathbf{T}\mathbf{b},\tag{18}$$

Let $\mathbf{TA} = \mathbf{M}_T - \mathbf{N}_T$ be the regular splittingof $\mathbf{TA}$, then the basic iterative method based on (18) can be defined as

$$\mathbf{x}^{(k+1)} = \mathbf{B}_T\mathbf{x}^{(k)} + \mathbf{c}_T, \tag{19}$$

where $\mathbf{c}_T = \mathbf{M}_T^{-1}\mathbf{b}_T$. The spectral condition number of the iteration matrix for the preconditioned system should be smaller than that for the original system. Thus, the matrix $\mathbf{T}$ should be constructed in such a way that it meets the above requirement and is easy to implement. We propose a simple and efficient preconditioning mechanism for the GS detection given as follows [41]:

$$\mathbf{T} = \mathbf{I} + \mathbf{R}, \tag{20}$$

where the nonsingular matrix $\mathbf{R}$ is defined by

$$\mathbf{R} = \begin{bmatrix} 0 & 0 & \dots & 0 \\ \vdots & \vdots & \ddots & \vdots \\ 0 & 0 & \dots & 0 \\ -a_{n1} & -a_{n2} & \dots -a_{n,m-1} & 0 \end{bmatrix}.$$

Thus, we obtain

$$\mathbf{A}_T = (\mathbf{I} + \mathbf{R})\mathbf{A}\mathbf{x} = (\mathbf{I} + \mathbf{R})(\mathbf{I} - \mathbf{L} - \mathbf{U})\mathbf{x} = (\mathbf{I} - \mathbf{L} - \mathbf{U} + \mathbf{R} - \mathbf{RL} - \mathbf{RU})\mathbf{x}, \tag{21}$$

wherever

$$\sum_{j=l+1}^{i} a_{lj}a_{jl} \neq 1, \quad l = 1, 2, \dots, i-1, \tag{22}$$

$(\mathbf{I} - \mathbf{L} + \mathbf{R} - \mathbf{RL} - \mathbf{RU})^{-1}$ exists, and for $\mathbf{A}_T$, the GS iteration matrix $\mathbf{B}_T$ is defined by

$$\mathbf{B}_T = (\mathbf{I} - \mathbf{L} + \mathbf{R} - \mathbf{RL} - \mathbf{RU})^{-1}\mathbf{U}. \tag{23}$$

For the preconditioned system, the nonsingular matrix $\mathbf{M} = \mathbf{I} - \mathbf{L} + \mathbf{R} - \mathbf{RL} - \mathbf{RU}$ and the matrix $\mathbf{N} = \mathbf{U}$. Then, the estimation of the transmitted signals based on the proposed preconditioning can be expressed as

$$\mathbf{x}_i^{(k)} = \frac{1}{\mathbf{A}_{Tii}}\left(\hat{\mathbf{y}}_{Ti} - \sum_{j<i}\mathbf{A}_{Tij}\mathbf{x}_j^{(k)} - \sum_{j>i}\mathbf{A}_{Tij}\mathbf{x}_j^{(k-1)}\right), \tag{24}$$

$$i, j = 1, 2, \dots, N_{iter}$$

where $\hat{\mathbf{y}}_T = (\mathbf{I} + \mathbf{R})\hat{\mathbf{y}}$. Algorithm 1 summarizes the proposed detector. The proposed detection algorithm arrives at the final convergence much faster compared to conventional iterative methods, which was verified through numerical simulations in the Results Section 4.

---

**Algorithm 1:** Proposed algorithm.

---

1 **Input**: $\mathbf{H}$, $\mathbf{y}$, M, K, $N_{iter}$, $E_x$, $\sigma^2$
2 **Preconditioning**:
3 $\mathbf{A} = \mathbf{H}^H \mathbf{H} + \frac{\sigma^2}{E_x} \mathbf{I_K}$
4 $\hat{\mathbf{y}} = \mathbf{H}^H \mathbf{y}$
5 $\mathbf{D} = diag(\mathbf{A})$
6 $\mathbf{s} = \mathbf{D}^{-1} \hat{\mathbf{y}}$
7 $\mathbf{R} = \mathbf{D}^{-1} \mathbf{A}$
8 $\mathbf{R}(1 : K - 1, :) = 0; \mathbf{R}(K, K) = 0$
9 $\mathbf{I}_K = 1 \times K$ identity matrix
10 $\mathbf{T} = \mathbf{I}_K + \mathbf{R}$
11 $\hat{\mathbf{y}}_T = \mathbf{Ts}$
12 $\mathbf{A}_T = \mathbf{TD}^{-1} \mathbf{A}$
13 **Initialization**:
14 $\hat{\lambda}_{max} = M \left( 1 + \sqrt{\frac{K}{M}} \right)^2$
15 $\mathbf{x}^{(0)} = \frac{1}{\hat{\lambda}_{max}} \hat{\mathbf{y}}$
16 **Iteration**:
17 **for** $k = 1, \ldots, N_{iter}$ **do**
18      **for** $n = 1, \ldots, K$ **do**
19          $\mathbf{x}_i^{(k)} = \frac{1}{\mathbf{A}_{Tii}} \left( \hat{\mathbf{y}}_{Ti} - \sum_{j<i} \mathbf{A}_{Tij} \mathbf{x}_j^{(k)} - \sum_{j>i} \mathbf{A}_{Tij} \mathbf{x}_j^{(k-1)} \right)$
20      **End for**
21 **End for**
22 **Output**: Detected signal, $\hat{\mathbf{x}}$;

---

## 4. Numerical Results

In this section, we evaluate the performance in terms of the symbol error rate (SER) of different detection algorithms in an uplink massive MIMO wireless communication system. To verify the validity of the proposed linear detection algorithm, we compared its performance with conventional GS [21] and preconditioned GS (CP-GS) [24] detectors. In addition, the proposed algorithm was also compared with state-of-the-art iterative detection methods such as Jacobi [25], Neumann series [17], second-order Richardson method (SORM) [27], and AOR [28]. The linear MMSE exact matrix inversion method was included as a benchmark. We assumed perfect knowledge of the channel state information is available at the receiver, and the channel matrices were generated using i.i.d. flat Rayleigh fading channel model. For a fair comparison, different system configurations employing higher-order modulation techniques such as 16-QAM and 64-QAM were considered. Table 1 summarizes the simulation model parameters.

**Table 1.** Summary of the simulated model parameters.

| Parameter | Value |
|---|---|
| Number of antennas at BS | $M \in \{128, 256\}$ |
| Number of users | $K \in \{16, 32, 64, 5{:}100\}$ |
| User antennas | Single-antenna users |
| SNR range | SNR $\in \{8{:}2{:}18\}$ dB |
| Average SNR per receive antenna | $KE_x/N_0$ |
| Number of realizations in the Monte Carlo simulations | $25 \times 10^3$ |
| Number of iterations for iterative detectors | 1 to 6 |
| Channel | MIMO |
| Channel model | Uncorrelated Rayleigh fading |
| Channel availability | Perfectly known at the receiver |
| Modulation type | 16-QAM, 64-QAM |
| Transmission | Uncoded |

### 4.1. Comparison of Different Initializers

In this subsection, the simulation results of the proposed algorithm employing the zero vector initial solution, diagonal initial solution, and proposed initial solution are provided. A massive MIMO system with 128 antennas at the BS and 16 users was considered. With the 64-QAM modulation technique, Figure 2 demonstrates that the proposed algorithm shows the worst performance with zero vector initialization. Moreover, it is observed that the proposed initialization reveals degraded detection results compared to the diagonal initial solution for $k = 1$ and $k = 2$. However, the developed algorithm with the proposed initializer shows a similar performance as that of the diagonal initialization for $k = 3$. Note that the diagonal initial solution requires relatively more computations. Hence, the proposed initial method is the best choice, which achieves the detection accuracy of the diagonal initial solution with reduced computations.

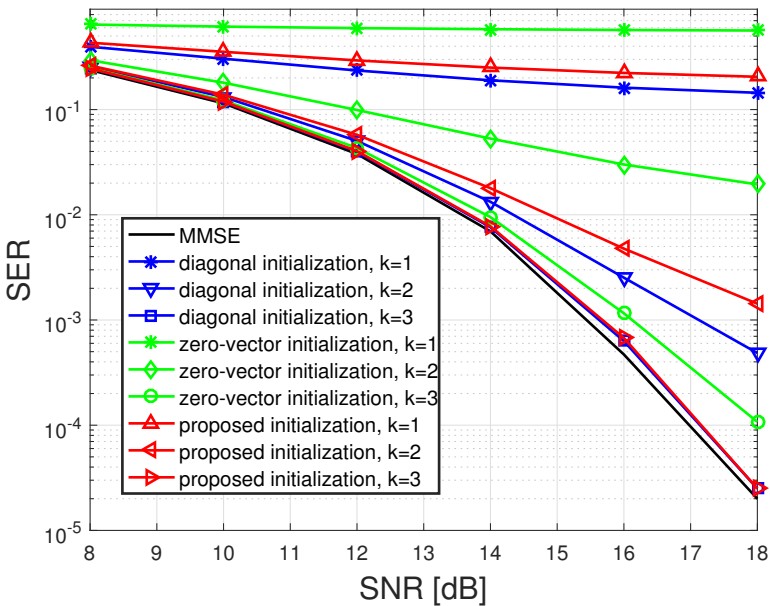

**Figure 2.** Performance of the proposed algorithm for a system deploying $M \times K = 128 \times 16$ with 64-QAM modulation scheme applying different initial methods.

### 4.2. Error Rate Performance

We first demonstrate the error performance of various detection techniques for a system with 256 antennas at the BS and 32 users in Figure 3. The 64-QAM modulation scheme was applied for this simulation. It can be easily observed from the plot that the performance of all iterative algorithms improved as the number of iterations increased. The CP-GS exhibited degraded performance compared to the conventional GS-based detector. Note, moreover, that the designed detection algorithm is superior to the aforementioned iterative schemes in terms of the error rate in the considered massive MIMO scenario. Furthermore, it achieved almost identical accuracy to that of the linear MMSE with $k = 4$.

In Figure 4, we increased the number of users and kept the same number of antennas at the BS as in Figure 3, to study the error rate performance of the proposed algorithm and the existing GS and CP-GS detection methods. The considered antenna configuration was $M \times K = 256 \times 64$ with the 16-QAM modulation technique. It can be clearly seen from the figure that all methods performed well for the given system settings. The GS performed better than CP-GS, and the main reason for the relatively better performance of GS is the diagonal initial solution. However, significantly better performance achieved by the proposed approach compared to the aforementioned detectors is clear from the plot. For the proposed algorithm, the SNR required to obtain an SER of $10^{-4}$ was 17.2 dB, whereas for the benchmark method, it was 17.03 dB. Thus, the performance deference between them was only 0.17 dB.

Next, we compared the proposed algorithm with state-of-the-art iterative detection methods to further validate the superiority of the designed approach. For an $M \times K = 128 \times 16$ massive MIMO system, Figure 5 reveals that the Jacobi- and Neumann-series-based detectors showed degraded performance. In contrast to Jacobi and Neumann, the AOR exhibited better performance. Note further that the detection performance of the SORM method improved with the number of iterations and achieved good results compared to Jacobi, Neumann, and AOR for the given iterations. However, the proposed algorithm achieved a lower error for the same number of iteration count than all aforementioned iterative methods. Moreover, it realized the MMSE performance with only four iterations.

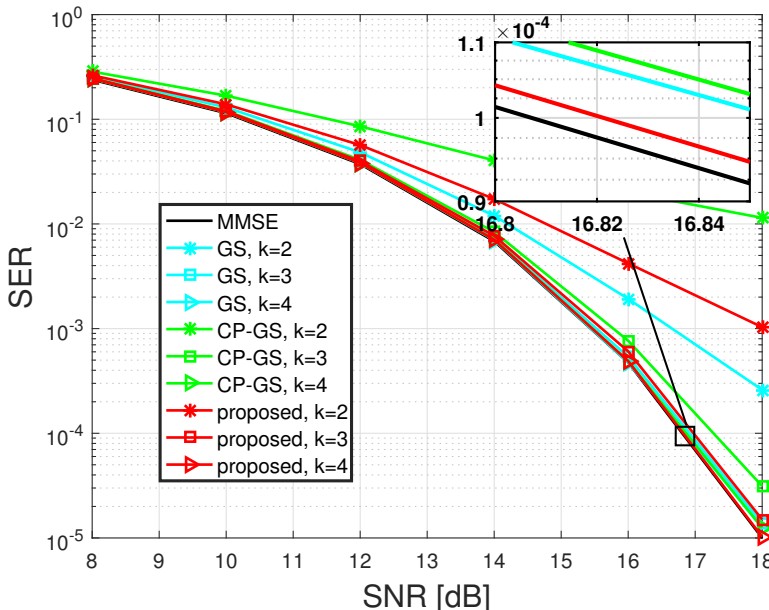

**Figure 3.** SER performance versus SNR of the proposed algorithm and GS- and CP-GS based algorithms for a $256 \times 32$ antenna system employing the 64-QAM modulation scheme.

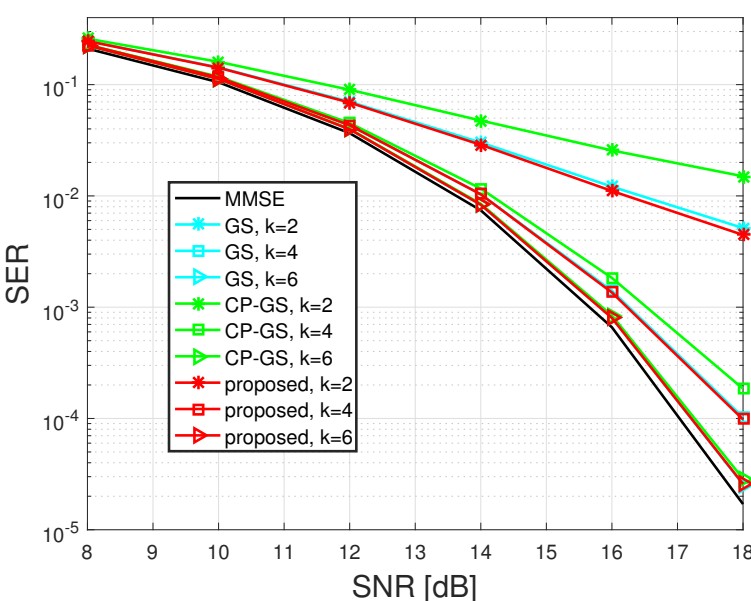

**Figure 4.** SER as a function of SNR for the systems deploying $M = 256$ antennas at the BS and 64 users with the 16-QAM modulation scheme.

In Figure 6, we study the SER performance of the proposed scheme as a function of the number of user terminals and compare it with recently reported GS- and CP-GS-based detection algorithms. In this case, the considered antennas at the BS were 128 and the 16-QAM modulation technique was employed. One can see that the the performance of the proposed algorithm, GS, and CP-GS was almost similar to that of the linear MMSE as the number of user terminals grew up to 25. However, for more than 25 antennas, there existed a gap between the iterative detectors and linear MMSE. The CP-GS exhibited degraded results compared to GS and the proposed algorithm. Further, it can be observed that the proposed detector converged faster than the conventional GS detector.

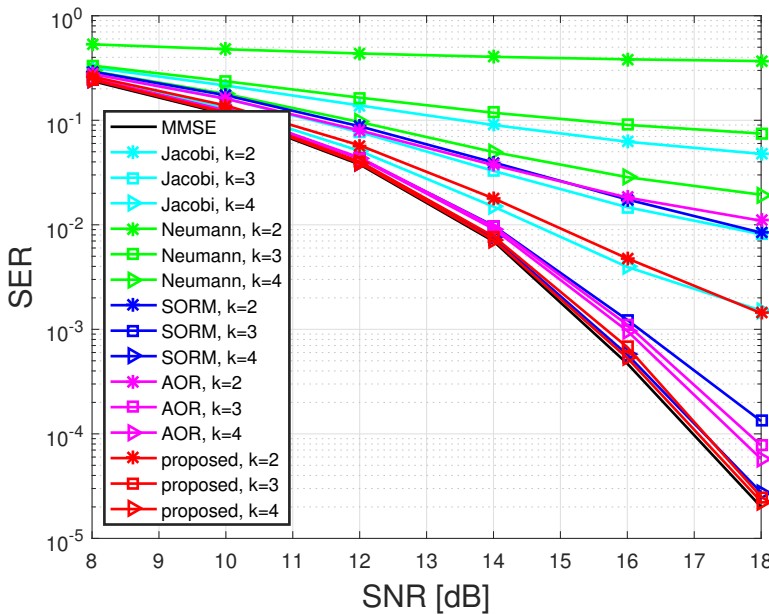

**Figure 5.** SER performance comparison of the proposed algorithm and other recently proposed methods for $M \times K = 128 \times 16$ antenna system employing 64-QAM modulation.

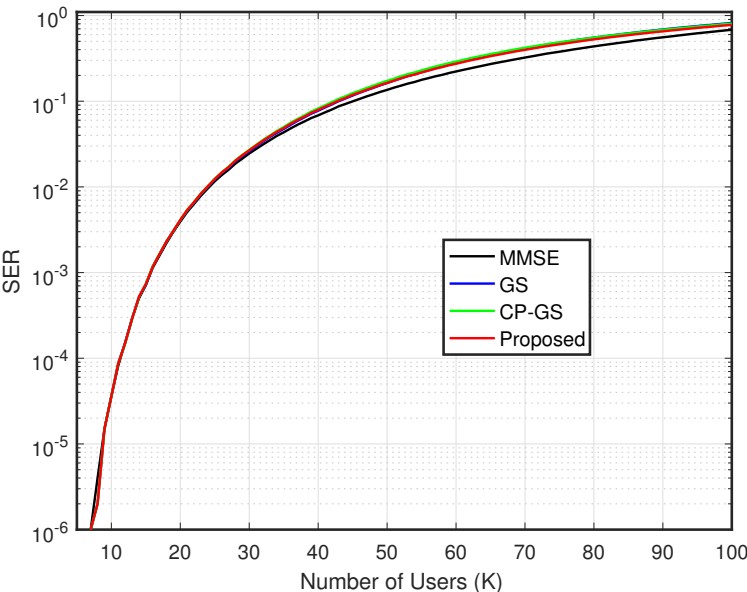

**Figure 6.** SER as a function of the number of users for the system deploying $Mr = 128$ antennas at the BS with the 16-QAM modulation technique.

To study the numerical stability of the proposed technique, we provide results of the error performance as a function of the number of iterations. Figure 7 demonstrates the results for $M = 128$ antennas at the BS with 16 user terminals utilizing the 64-QAM modulation scheme. For various SNR values, the figure shows that, after a few number of iterations, the performance became stable. We can observe that, to attain stability for smaller SNR values, a smaller number of iterations is required. Thus, the proposed method is numerically stable and a few iterations are sufficient to achieve the desired performance.

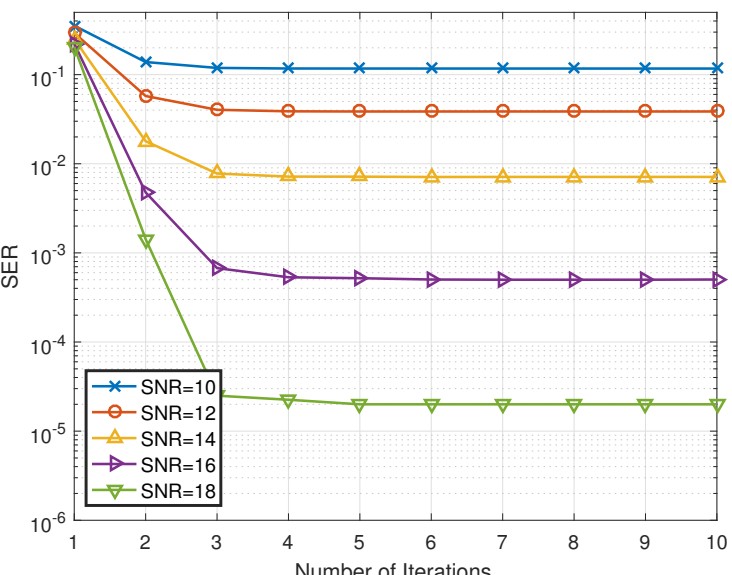

**Figure 7.** SER versus the number of iterations of the proposed detector at different values of SNR for $M \times K = 128 \times 16$ massive MIMO system employing the 64-QAM modulation scheme.

Figure 8 shows the simulation results for the SER performance against the number of iterations. In this case, we compared the proposed technique with other state-of-the-art techniques for a system with 128 antennas at the BS and 32 users employing 16-QAM modulation with the SNR set to 16 dB. It can be noted that the Jacobi and Neumann methods achieved a high error floor. The main reason that the Jacobi method attained a high error floor

is the damping factor, which is only applicable for a certain antenna scenario. Further observe that the AOR also showed slow convergence, and its error performance was degraded in this case, which is due to its sensitivity to acceleration and relaxation parameters. The convergence rate of the GS-based detectors was better than all other iterative detectors. It can be observed that the proposed algorithm and the conventional GS obtained almost the same performance for higher iterations, and the proposed algorithm had a better SER than GS up to three iterations. However, in all above-provided performance results, the proposed method exhibited the fastest convergence compared to all mentioned iterative methods and achieved desired results only within a few iterations. Thus, it can be concluded that the proposed detector is superior to all compared iterative detectors in terms of convergence and error performance.

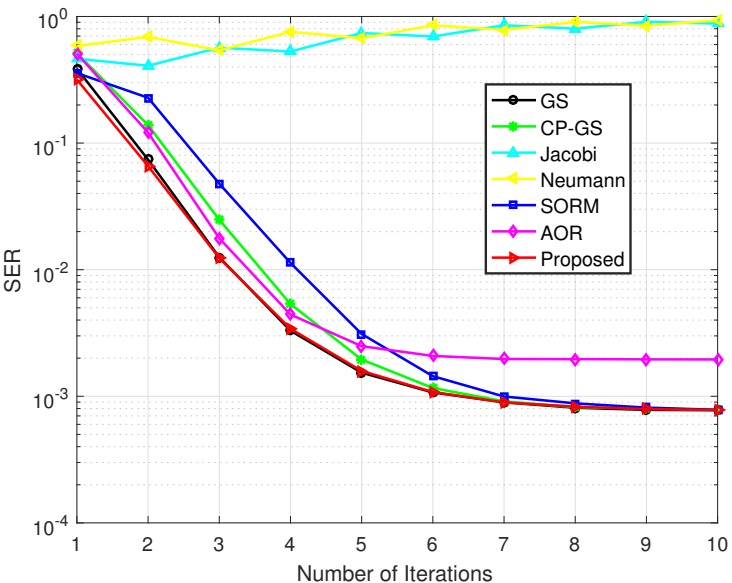

**Figure 8.** SER performance as a function of the number of iterations for $M \times K = 128 \times 32$ massive MIMO system at 16 dB SNR using 16-QAM modulation.

### 4.3. Complexity Analysis and Comparison

The computational complexity required for estimating the signal in terms of required number of multiplications is analyzed in this section. We first computed the complexity involved in each step of the proposed detection approach, then compared it with the conventional GS-based detectors. Since the complexity of $\mathbf{A}$ and $\hat{\mathbf{y}}$ is required by all methods, we calculated the complexity of the later parts. The complexity of iterative methods mainly depends on the iteration cycles. One can see from (24) that $K - 1$ total multiplications were required to obtain $\mathbf{x}_i^{(k)}$ for each $i$ and $k$. While there were $K$ number of elements in vector $\mathbf{x}^{(k)}$, the overall required complex multiplications for the GS iteration were $i(K^2 - K)$. The computations of the initializer originate from solving (13). It can be easily found that it involves $K + 3$ multiplications to achieve $\mathbf{x}^{(0)}$. Finally, we calculated the complexity involved in the preconditioning step. For this step, first, we needed to compute $\mathbf{x}$ and $\mathbf{R}$, that is Lines 6 and 7 in Algorithm 1. It can be easily observed that $K^2 + K$ multiplications are needed to obtain $\mathbf{R}$. Similarly, $K$ multiplications are required to obtain vector $\mathbf{x}$. Next, it is required to compute $\hat{\mathbf{y}}_T$ and $\mathbf{A}_T$, that is Steps 10 and 11 in Algorithm 1. To compute $\hat{\mathbf{y}}_T$, it involves the multiplication of $\mathbf{T}$ and vector $\mathbf{x}$. Since we used only $K - 1$ elements of the last row of matrix $\mathbf{T}$ in the proposed preconditioning, it requires $K - 1$ multiplications to obtain $\hat{\mathbf{y}}_T$. The computation of $\mathbf{A}_T$ involves a multiplication matrix $\mathbf{T}$ and matrix $\mathbf{R}$. Although, it involves the multiplication of matrices. as mentioned above, $K - 1$ elements of $\mathbf{T}$ were used. Therefore, its computational complexity is $K^2 - K$. Thus, the total complexity of the proposed algorithm for each iteration is $k(K^2 - K) + 2(K^2 + K) - 1$. Note

that the proposed detector has one-order-less computational complexity than the linear MMSE. Next, we compared the complexity of the proposed method with recently reported iterative methods.

First, the complexity of the proposed algorithm and the GS-based detection schemes is compared in Figure 9. The complexity of the benchmark linear MMSE is also included. The figure shows that the linear MMSE has the highest complexity (cubic of the number of users) and the conventional GS detector has the lowest complexity among all detectors. The proposed algorithm and CP-GS exhibit similar complexity, as can be seen from the plot. However, the convergence rate of the proposed algorithm is much faster than the GS- and CP-GS-based detectors, which has already been demonstrated. Thus, the proposed algorithm requires less iterations to achieve the MMSE performance compared to other methods, which ultimately reduces the number of computations of the proposed detector. In addition, GS uses the diagonal-matrix-based initial solution, which has higher complexity than the proposed approximate-eigenvalue-based initial solution.

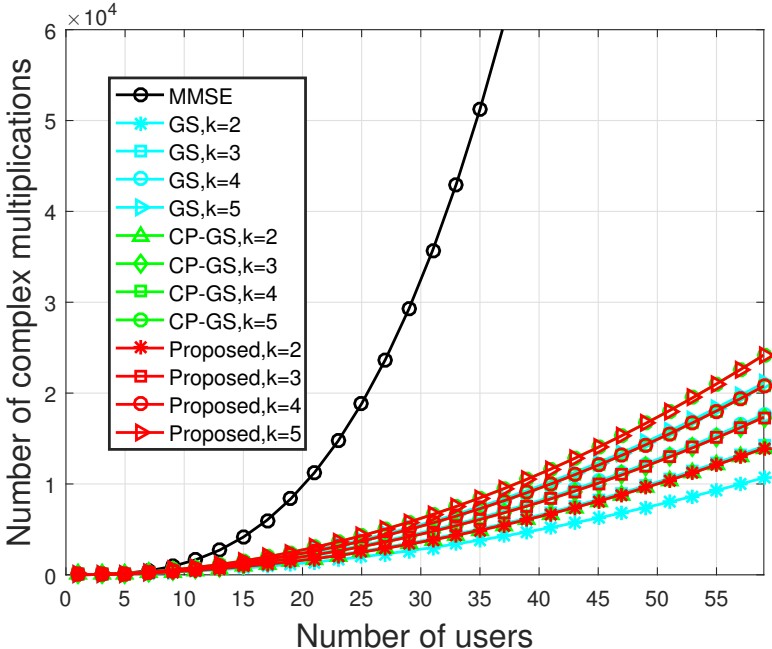

**Figure 9.** Complexity comparison of various detection algorithms against the number of users.

In addition, the computational complexity of state-of-the-art detectors is also provided in Table 2. The complexity of the Neumann series is included for $k \geq 3$, since it usually needs more than three iterations to achieve the desired performance. It can be seen that it has cubic complexity for $k \geq 3$, which is very high for massive MIMO systems. The complexity of the proposed algorithm is slightly higher than other state-of-the-art detectors such as Jacobi, SORM, and AOR. However, the proposed detector performs much better compared to other methods, which was shown in the previous subsections. Furthermore, it obtains the desired detection results with fewer iterations, whereas the aforementioned detection schemes need more iterations for obtaining the same performance, which further reduces the computations of the proposed algorithm. In summary, the proposed detector can obtain the best trade-off between the computational complexity and error rate performance among the discussed iterative detection schemes.

**Table 2.** Computational complexity.

| Detector | Complexity |
| --- | --- |
| GS [21] | $(k+1)K^2 + 4K$ |
| CP-GS [24] | $(k+2)K^2 + (2-k)K - 1$ |
| Jacobi [25] | $k(2K^2 - K) + 2K$ |
| Neumann Series [17] | $(k-2)K^3 + 3K^2 \quad (k \geq 3)$ |
| SORM [27] | $k(K^2 + 2K + 7)$ |
| AOR [28] | $k(3K^2 + 7K)/2$ |
| Proposed | $k(K^2 - K) + 2(K^2 + K) - 1$ |

## 5. Conclusions

We considered an approximated linear MMSE detection for massive MIMO uplink systems. We presented a low-complexity matrix-inverse-free signal detection algorithm based on the GS method. By taking full advantage of the special property of massive MIMO systems, a novel initial method was proposed and then approximated. It was shown that the new initializer realizes the same detection results to that of the conventional diagonal initial solution with decreased computational burden. Moreover, it significantly outperforms the traditional zero vector initializer. To approach the MMSE accuracy with a small number of iterations, an effective preconditioned method was designed. The complexity–performance tradeoff of the proposed detector was also illustrated and compared with the recently reported detection schemes. The analysis and simulation results demonstrated that the proposed method, although having a much lower computational complexity, can achieve similar SER performance as the linear MMSE and obtains a better performance compared to existing iterative detectors.

**Author Contributions:** Methodology, M.A.; Software, M.A. and I.A.K.; Validation, X.Z.; Formal analysis, M.A. and X.S.; Investigation, M.A.; Resources, X.Z., X.S. and Y.Q.; Writing—original draft, M.A.; Writing—review & editing, I.A.K.; Visualization, Y.Q.; Supervision, X.Z.; Funding acquisition, X.Z. All authors have read and agreed to the published version of the manuscript.

**Funding:** This work is supported by the National Natural Science Foundation of China (62101250), the Natural Science Foundation of Jiangsu Province (BK20210281), the Jiangsu Key Research and Development Project (BE2020101), the National Key Research and Development Project Grant (2020YFB1807602), the National science foundation of China (61971217, 61971218, 61631020, 61601167), the Jiangsu NSF Grant (BK20200444), the Jiangsu Planned Projects for Postdoctoral Research Funds (2020Z013), the Postgraduate Research & Practice Innovation Program of Jiangsu Province (KYCX21_0215), and the China Postdoctoral Science Foundation (2020M681585).

**Data Availability Statement:** The data that support the findings of this study are available from the corresponding author upon reasonable request.

**Conflicts of Interest:** The authors declare no conflict of interest.

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
