# Peer review of "High-Precision Iterative Preconditioned Gauss–Seidel Detection Algorithm for Massive MIMO Systems"

_electronics, doi:10.3390/electronics11223806_

Round 1

Reviewer 1 Report

Title: High-Precision Iterative Preconditioned Gauss-Seidel Detection Algorithm for Massive MIMO Systems

The paper predominantly focuses on the reduction of the complexity in the algorithms applied in massive MIMO systems that employs large number of antennas in the transmitter and the receivers. Specifically, it uses the Gauss-Siedal method proposes two changes one at initialization and other for preconditioning. This methods approaches MMSE performances when number of receivers are more than 2 (which will be the case in general). The proposed solution seems promising. However, some queries need to be resolved and some modifications are required before it can be accepted:

1) Please provide a detail on the simulation model as the results have not been validated using experiments. Perhaps a figure that summarizes all system/channel modeling related things.

2)Please provide comparison with other works that endeavor to achieve such complexity reduction. A comparison table would be highly appreciated.

3) Section 2:  Please provide a figure for the system model and show the respective receivers and transmitter. It be be more comprehensible.

4) Please define x variables for equation (7) again.

5) Fig. 6 : Authors can use different colors to further demarcate different curves at varying SNRs

Author Response

Please see the attachment word file, named (Response-to-Reviewer_1)

Reviewer 2 Report

This paper studies the iterative preconditioned Gauss-Seidel detection algorithm for massive MIMO systems, where a initialization is proposed. Some numerical experiments are given to evaluate the performance. However, there are some issues with respect to system model, algorithm description, and discussion. Some detailed comments are as follows.

1. The use of symbols in the manuscript is confusing, and some symbols are not defined in detail, which makes it difficult for readers to understand.

- For example, page 2, ll.109, xm, page 2, ll.125,Ex, etc.

- In page 6, the authors mentioned that all elements an,m of R are non-zero, which is contradictory to the expression for R.

- Eq.(18), x is missing.

- Eq.(19),k is not defined.

2. The authors claim the the maximum eigenvalue of A is approximated by (11). According to (3) and (8), the maximum eigenvalue of A should be M+sigma^2/Es, please explain it.

3. In section 3.2, the algorithm is not clearly described.

4. The authors claim that preconditioning improves the spectral properties of the coefficient matrix, thereby improving the convergence of iterative method, please classify it.

5. In the simulation experiment, the convergence analysis and comparison of several mentioned algorithms should be given.

Author Response

Please see the attachment word file, named (Response-to-Reviewer_2)

Reviewer 3 Report

The paper clearly gives the environment, assumptions, requirements and objectives of the problem in hand, and points out major issues or difficulties when dealing with the problem and the system design. In the paper, the performance evaluation is also presented, which makes the paper a complete work. The paper is well-organized, but it can be improved as follows:

1.       This paper is based on a few strong assumptions, which could be critical to the study. It would help if the authors give a detailed description on these assumptions/models.

2.       A vest amount of existing studies has proposed the similar concept. It would be better to provide a comprehensive performance comparison. Without this work, the reviewer cannot identify the contribution of this paper. The Introduction (Section I) and References Sections are not satisfactory. For instance:

[R1] P. Seidel, D. Gregorek, S. Paul and J. Rust, "Efficient Initialization of Iterative Linear Massive MIMO Uplink Detectors by Binary Jacobi Synthesis," WSA 2019; 23rd International ITG Workshop on Smart Antennas, 2019, pp. 1-5.

3.       The system implementation can be explained better since the main descriptions are not clear for the readers. The organization and presentation can be improved.

Author Response

Please see the attachment word file, named (Response-to-Reviewer_3)